# Interpretable ensemble machine learning framework for cardiovascular disease prediction using EMR data and large language models in Ethiopia

Alemu Kumilachew Tegegnie ⓘ *, Kibrom Tewolde

Faculty of Computing, Bahir Dar University Institute of Technology, Bahir Dar, Ethiopia

* alemupilatose@gmail.com

## Abstract

Cardiovascular diseases (CVDs) are leading causes of morbidity and mortality globally, with a growing burden in low- and middle-income countries such as Ethiopia. Early detection is limited by resource constraints, low screening uptake, and a lack of predictive tools tailored to local healthcare systems. This study presents an interpretable ensemble machine learning framework for predicting CVD risk via structured electronic medical record (EMR) data from public hospitals in Addis Ababa. We trained an XGBoost classifier on 20,960 anonymized records containing demographic, clinical, and physiological attributes. Preprocessing involves handling missing values, outlier capping, one-hot encoding, rare-category grouping, and dimensionality reduction. SHapley additive explanations (SHAPs) were used for feature attribution, and a large language model (Gemini) was used to translate SHAP outputs into plain-language narratives to enhance interpretability. The model achieved an accuracy of 0.99, with strong precision (0.99), recall (0.98), and F1-scores across both classes. SHAP analysis identified general_plan, history of present illness (HPI), musculoskeletal system (MSS) and diagnosis as key predictors. The integration of SHAP and LLMs provided transparent, clinician-friendly insights into model outputs, supporting adoption in resource-limited settings. This study demonstrates that combining ensemble learning with explainability techniques can yield highly accurate and interpretable CVD prediction models, offering potential for integration into clinical decision-support systems in Ethiopia.

## 1. Introduction

Cardiovascular diseases (CVDs) account for approximately 17.9 million deaths annually, representing 32% of all deaths worldwide [1]. In Ethiopia, urbanization, demographic shifts, and lifestyle transitions have contributed to a growing CVD burden, particularly in cities such as Addis Ababa [2]. However, limited diagnostic

**Data availability statement:** The dataset supporting the findings of this study is publicly available. The original dataset is hosted on Figshare and remains fully functional at https://doi.org/10.6084/m9.figshare.30436426. In addition, an updated version of the dataset is available via GitHub at https://github.com/ale-mupilatose/My_Dataset1/commit/03372e442e-112aa892403fa08b37dc529091be1f. Both links provide open access to the data without restrictions.

**Funding:** The author(s) received no specific funding for this work.

**Competing interests:** NO authors have competing interests.

**Abbreviations:** AI: Artificial Intelligence, CVD: Cardiovascular Disease, DL: Deep Learning, EMR: Electronic Medical Record, IQR: Interquartile Range, LLM: Large Language Model, ML: Machine Learning, MSS: Musculoskeletal System, SHAP: SHapley Additive exPlanations, XAI: Explainable Artificial Intelligence, XGBoost Extreme Gradient Boosting

infrastructure, delayed care-seeking behavior, and insufficient screening programs hinder early detection and timely intervention [3,4]. These challenges are further compounded by resource constraints and the lack of localized clinical decision-support tools, leaving clinicians without effective means to identify at-risk individuals early in the disease trajectory.

Machine learning (ML) offers significant promise in improving early diagnosis and risk prediction of CVDs by leveraging complex, high-dimensional data patterns [5]. Nevertheless, the adoption of ML models in Ethiopian clinical practice has been limited, largely due to a lack of interpretability—most models function as "black boxes," making it difficult for healthcare professionals to understand and trust their outputs [6]. This barrier is particularly critical in resource-limited settings, where transparency and simplicity are essential for frontline adoption.

This study addresses these gaps by developing a highly interpretable ensemble machine learning model for CVD prediction using structured electronic medical record (EMR) data from public hospitals in Addis Ababa. A unique aspect of this work is the use of SHapley Additive exPlanations (SHAPs) to identify and quantify the contributions of clinical, demographic, and physiological features such as diagnosis, general care plans, and musculoskeletal system (MSS) indicators. To further enhance transparency, a large language model (LLM) was employed to translate SHAP values into natural-language narratives, making the model's outputs more accessible and actionable for clinicians without technical expertise.

By grounding the model in real-world EMR data from over 20,000 anonymized patient records and enhancing interpretability through SHAP and LLM integration, this work introduces a novel, clinician-friendly framework tailored to Ethiopia's health-care context. This demonstrates how explainable ensemble learning can support evidence-based decision-making in low-resource environments. These contributions place the study at the intersection of machine learning, healthcare informatics, and explainable AI, offering a scalable approach to improving clinical risk prediction and transparency in data-driven health systems.

## 2. Related works

CVD is now defined as "presence of a documented cardiovascular diagnosis in the EMR (coded as 1) and absence as (0). Machine learning (ML) and deep learning (DL) techniques have been widely applied in cardiovascular disease (CVD) prediction, offering promising improvements in early diagnosis and risk stratification through data-driven approaches. Traditional statistical models such as logistic regression and decision trees have been commonly employed for CVD risk estimation [7]. However, these methods often struggle to capture the complex, nonlinear relationships inherent in clinical and physiological data. More advanced ensemble learning techniques, particularly XGBoost, have demonstrated superior predictive performance by effectively handling high-dimensional datasets and providing interpretable feature importance measures [8,9]. The challenge of model interpretability in clinical settings has led to the growing adoption of explainable artificial intelligence (XAI) techniques. SHapley Additive exPlanations (SHAP) is one such method that enables

granular interpretation of feature contributions, helping clinicians understand the rationale behind model predictions and fostering trust in automated decision-support tools [6]. Several studies integrating SHAP with CVD prediction models report enhanced usability and acceptance among healthcare professionals, demonstrating that explainability is crucial for bridging the gap between complex models and clinical practice [10,11]. In parallel, large language models (LLMs) have emerged as powerful tools for generating natural-language explanations from structured model outputs. By translating numerical or statistical interpretations into clinician-friendly narratives, LLMs can further improve the transparency and accessibility of AI-driven predictions. Recent work has explored the integration of LLM-generated narratives with predictive models to enhance clinical adoption and communication [12]. Despite these advances, few studies have applied such combined frameworks in low-resource healthcare environments, where electronic medical records (EMRs) are often incomplete, inconsistently structured, or underutilized, as is the case in Ethiopia.

While considerable progress has been made in applying ML for CVD prediction globally, several critical gaps limit the applicability of existing models to the Ethiopian context. Most CVD prediction models are trained and validated on datasets from high-income countries. These datasets reflect healthcare systems, patient demographics, and disease profiles that differ markedly from those found in Ethiopia. Ethiopian patient populations exhibit unique clinical and demographic characteristics shaped by local genetics, environmental conditions, healthcare-seeking behaviors, and cultural factors. These differences influence disease manifestation and progression, rendering externally developed models less reliable or potentially biased when applied without adaptation. Using models trained on non-local data without appropriate calibration risks inaccurate risk estimation and biased predictions. Such models may fail to account for local comorbidities, prevalent lifestyle factors, or variations in diagnostic practices. Moreover, resource constraints in Ethiopian healthcare—such as limited access to diagnostic tools and inconsistent data recording—demand models sensitive to these operational realities. Without tailoring, predictive tools may lack clinical relevance and fail to support timely and effective interventions. Transparency and explainability are particularly important in resource-limited settings, where healthcare providers may have limited training in advanced analytics. Models functioning as "black boxes" offer little actionable insight, which can hinder their acceptance and use. Hence, developing interpretable models that provide clear, clinician-friendly explanations is critical for facilitating trust and integration into Ethiopian clinical workflows.

By grounding model development in Ethiopian EMR data and combining XGBoost with SHAP-based interpretability and Gemini LLM-generated narrative explanations, this study aims to fill these gaps. To the best of our knowledge, this represents one of the first efforts to develop a high-performance, interpretable, and contextually relevant CVD prediction framework tailored specifically to the Ethiopian healthcare setting. This approach not only advances methodological innovation but also addresses the practical challenges of applying AI in low-resource environments, contributing valuable insights to the field of AI-driven cardiovascular risk stratification.

## 3. Methods

### 3.1. Study design and data source

This study employed a retrospective, data-driven research design using anonymized electronic medical records (EMRs) collected from public hospitals under the Addis Ababa Health Bureau. The anonymized dataset, comprising 20,960 patient records, was collected and accessed for research purposes on 23/02/2025. Each record contained demographic details, vital signs, system-based clinical examinations, diagnostic assessments, and planned interventions. The outcome variable was binary, representing the presence (1) or absence (0) of CVD. Records were included if complete demographic and diagnosis data were available; collected on 23/02/2025.

### 3.2. Data acquisition and clinical definitions

In this study, Cardiovascular Disease (CVD) was defined based on formal clinician diagnoses documented within the EMR system, utilizing ICD-10 (International Classification of Diseases, 10th Revision) standards. Clinical predictors such

as the Musculoskeletal System (MSS), Neurological System (NEURO), and Respiratory System (RESP_SYST) represent assessment modules within the EMR that capture findings from the 'Review of Systems' and physical examinations. These clinical procedures follow the Ethiopian Primary Health Care Clinical Guidelines [13] and EMR clinical documentation protocols and usage [14], ensuring that definitions are grounded in professional clinician assessment rather than patient self-reporting.

### 3.3. Dataset description and selected features

The dataset initially included 29 mixed-type features, encompassing continuous, categorical, and text-based variables. After preprocessing, feature expansion via one-hot encoding resulted in 93,884 columns, which were then reduced to 489 columns by aggregating rare categories (<1% frequency) to improve computational efficiency.

The dataset includes the following key features:

- **Demographics:** Age, sex, residence

- **Physiological measures:** Pulse rate, respiratory rate, body temperature

- **Clinical system evaluations:** Musculoskeletal system (MSS), neurological system (NEUR_SYST), cardiovascular system, respiratory system

- **Patient history:** History of present illness (HPI), past diagnoses

- **Care plans:** General plan, planned interventions

- **Other features:** Duration of illness, chief complaints, and additional categorical indicators

Tables 1–3 summarize the preprocessing steps, feature types, transformations, and the class distribution of the target variable:

Table 2 summarizes the dataset pre-processing stages. It shows how the original dataset with 29 mixed-type features was expanded to 93,884 columns after one-hot encoding, then significantly reduced to 489 columns by grouping rare categories to improve computational efficiency.

**Table 1. Summary of data preprocessing steps.**

| Step | Description |
|---|---|
| Missing value handling | Median imputation; excluded fully missing *weight* |
| Outlier management | IQR capping for *pulse*, *respiratory rate*, *duration of illness* |
| Categorical encoding | One-hot encoding; rare-category grouping (<1%) |
| Feature scaling | Z score normalization (StandardScaler) |
| Dimensionality reduction | Removal of redundant and low-variance features |

**Table 2. Pre-processing summary of the dataset.**

| Stage | Rows | Columns | Description |
|---|---|---|---|
| Raw dataset | 20,960 | 29 | Mixed categorical, text, and numeric features |
| After one-hot encoding | 20,960 | 93,884 | Expanded due to categorical encoding |
| After rare-category grouping | 20,960 | 489 | Reduced dimensionality for computational efficiency |

**Table 3. Class distribution.**

| Class | Count |
|---|---|
| CVD present (1) | 10,710 |
| CVD absent (0) | 10,219 |

Table 3 presents the class distribution of the target variable. It shows a relatively balanced dataset, with 10,710 instances labeled as CVD present (1) and 10,219 instances labeled as CVD absent (0). This balance suggests that the model is unlikely to suffer from severe class imbalance issues during training and evaluation.

### 3.4. Preprocessing and imputation

Data preprocessing aims to increase quality, reduce noise, and optimize the dataset for machine learning models.

- Handling missing values: Missing values in continuous physiological variables (e.g., age, pulse rate) were handled using median imputation. Median imputation was selected for its robustness against outliers common in clinical datasets. However, it is recognized that this method may underestimate data variability. For categorical variables where documentation was sparse, missingness was treated as a distinct category ('Not Recorded') to avoid introducing synthetic bias into the clinical narratives. For enhanced statistical rigor, we acknowledge that Multiple Imputation (MI) serves as a more principled approach to handling missing data in clinical outcomes and predictors. Future iterations of this framework will incorporate MI to validate the stability of predictive associations across varying missingness mechanisms.

- Outlier management: Extreme values for *pulse rate*, *respiratory rate*, and *duration of illness* were adjusted via the interquartile range (IQR) capping method, where values below $Q1 - 1.5 \times IQR$ or above $Q3 + 1.5 \times IQR$ were replaced with the respective lower or upper bounds, thereby reducing the impact of extreme values while retaining all data.

- Categorical encoding: Categorical variables were transformed via one-hot encoding. High-cardinality variables were reduced by aggregating rare categories (occurring in <1% of records) into an "Other" category.

- Feature scaling: Continuous features (*pulse rate*, *respiratory rate*, *duration of illness*, and *temperature*) were standardized via z score normalization through StandardScaler, resulting in zero-centered variables with unit variance.

- Dimensionality reduction: Low-variance and redundant features were removed to minimize dimensionality, reduce the computational load, and improve model generalizability. This step reduces noise, computational load, and risk of overfitting, thereby improving model generalizability.

### 3.5. Feature selection strategy

To ensure both clinical relevance and statistical robustness, a feature selection strategy combining expert knowledge and preliminary statistical analysis and relevance assessment was applied. This combined approach ensured that the model leveraged clinically meaningful and statistically informative features for robust prediction.

1. **Expert Knowledge-Based Preselection:**

Clinical professionals from the Addis Ababa Health Bureau were consulted to identify features with known relevance to CVD risk and patient status. High-priority variables included age, sex, pulse rate, HPI, MSS, general care plans, and documented diagnoses.

2. **Preliminary Statistical Analysis and Relevance Assessment:**

Before implementing the ensemble machine learning framework, a rigorous preliminary analysis was conducted to examine the data structure and the associations between predictors and the CVD outcome. Bivariate analysis was performed

to evaluate these relationships: categorical variables (e.g., HPI, General Plan) were assessed using the Pearson Chi-square ($X^2$) test, while continuous variables were compared using independent t-tests, depending on the normality of their distribution.

Following the assessment of associations, the feature set was further refined through statistical relevance filters. Features exhibiting extremely low variance or negligible predictive potential following one-hot encoding were removed to reduce dimensionality and computational noise. Additionally, correlation analysis was executed to identify and mitigate multi-collinearity among continuous features. This integrated approach ensures that the underlying clinical distribution is transparent and that the feature importance subsequently identified by the XGBoost and SHAP models is both statistically robust and clinically grounded

## 3.6. Model development and prediction

Cardiovascular disease (CVD) risk was predicted using an XGBoost (Extreme Gradient Boosting) classifier, chosen for its efficiency, regularization capability, and ability to handle high-dimensional structured data. The model was trained on 20,960 anonymized EMR records containing demographic, clinical, and physiological features. Hyperparameters were optimized through iterative experimentation: learning rate = 0.1, maximum tree depth = 6, number of estimators = 100, and L2 regularization ($\lambda = 1$). The dataset was split into 80% training and 20% testing subsets, with 10-fold cross-validation applied on the training set to ensure robustness and prevent overfitting. This approach enabled accurate and reliable CVD prediction while maintaining model generalizability.

## 3.7. Explainability layer

To enhance interpretability, SHapley additive exPlanations (SHAPs) were applied to quantify the contribution of each feature to model predictions at both the global and individual levels.

Additionally, large language models (LLMs) (Gemini) were integrated to generate plain-language narratives summarizing SHAP outputs. These narratives aimed to improve accessibility for clinicians and non-technical stakeholders, facilitating practical integration into decision-making workflows.

## 3.8. Evaluation metrics

Model performance was evaluated using:

$$\textbf{Accuracy:} \ (TP + TN) / (TP + TN + FP + FN)$$

$$\textbf{Precision:} \ TP / (TP + FP)$$

$$\textbf{Recall} \ (\textbf{Sensitivity}) \textbf{:} \ TP / (TP + FN)$$

$$\textbf{F1-score:} \ 2 \times (Precision \times Recall) / (Precision + Recall)$$

Additionally, confusion matrices were generated to visualize the classification performance. A model was considered satisfactory if its accuracy, precision, recall, and F1 score exceeded 90% across both classes.

## 3.9. Ethics statement

This study utilized secondary, anonymized electronic medical record (EMR) data obtained from the Addis Ababa Health Bureau, the regulatory authority overseeing all public hospitals in Addis Ababa, Ethiopia. As the data were de-identified and collected for administrative and clinical purposes, no direct patient interaction occurred. Accordingly, the requirement

for individual informed consent and formal ethics approval was waived by the Institutional Review Board (IRB) of Bahir Dar University Institute of Technology, in line with the Ethiopian National Research Ethics Review Guideline. This waiver also complies with national data protection regulations and the ethical principles outlined in the Declaration of Helsinki (2008 revision). Data access was granted through official channels and used exclusively for academic research purposes.

## 4. Results

### 4.1. Model performance

The XGBoost model demonstrated exceptional predictive ability in identifying the presence of cardiovascular disease (CVD). The model achieved an overall accuracy of 0.99, with precision and recall values of 0.99 and 0.98, respectively, resulting in an F1 score of 0.99. These metrics underscore the robustness of the model in correctly classifying both CVD presence and absence cases, as detailed in Table 4.

### 4.2. Confusion matrix

Table 5 presents the confusion matrix of the XGBoost model predictions. The matrix reveals minimal false positives and false negatives, with 2034 true negatives and 2103 true positives, indicating excellent discriminative power and reliability in prediction.

### 4.3. Learning curves

The training and validation accuracy and loss curves, shown in Fig 1, exhibit stable convergence following the application of regularization techniques. This stability suggests that the model effectively mitigates overfitting, thereby enhancing generalizability to unseen data.

### 4.4. Feature importance (SHAP)

Fig 2 shows the SHAP summary plot, which identifies the key features contributing to model predictions. Notably, variables such as the general plan, history of present illness (HPI), musculoskeletal system (mss), and diagnosis emerged as the most influential predictors of cardiovascular disease risk.

The SHAP summary plot in Fig 3 shows that among the 27 original categorical features, only a few have a strong influence on the model's predictions. Specifically, general_plan, hpi, and diagnosis have the greatest impacts, with both low (blue) and high (red) values significantly affecting the output. Features such as mss, temp, and chief_compl have moderate impacts, whereas the remaining features, including dur_illness, neur_syst, and gus, contribute minimally. Overall, the model's predictions are driven primarily by a small subset of influential features.

**Table 4. Performance metrics.**

| Metric | CVD Present | CVD Absent |
|---|---|---|
| Precision | 0.99 | 0.99 |
| Recall | 0.98 | 0.99 |
| F1-score | 0.99 | 0.99 |

**Table 5. Confusion matrix for XGBoost predictions.**

| | Predicted 0 | Predicted 1 |
|---|---|---|
| Actual 0 | 2034 | 10 |
| Actual 1 | 39 | 2103 |

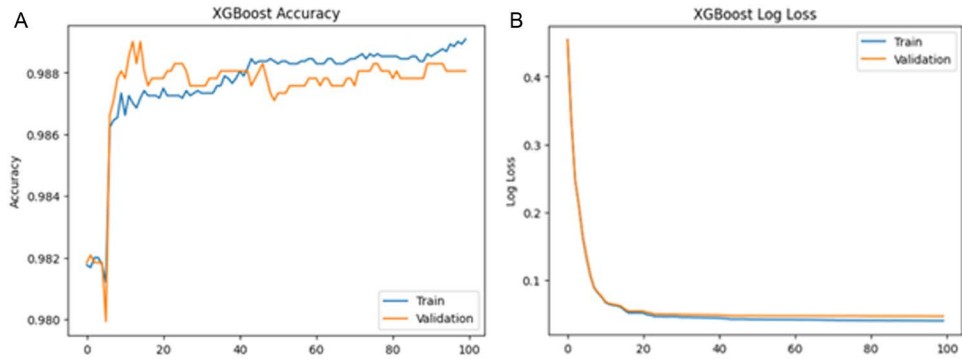

**Fig 1.** (a). Validation accuracy for XGBoost. (b). Validation Loss for XGBoost.

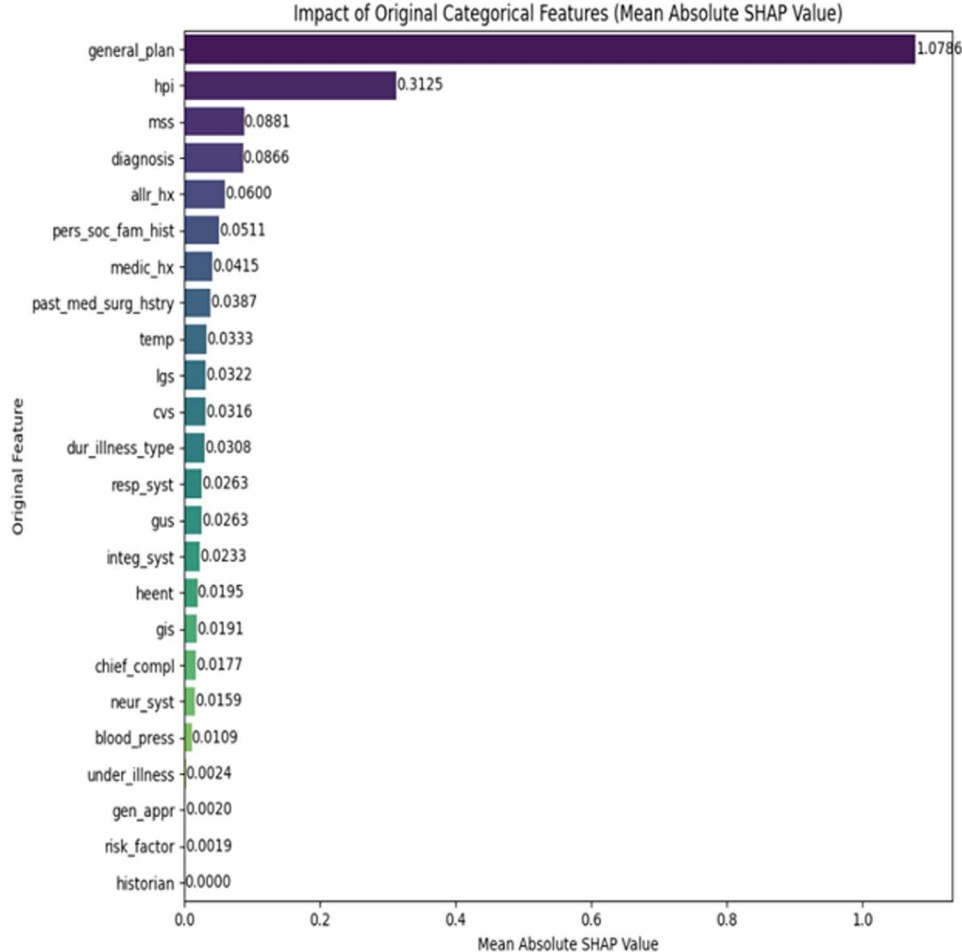

**Fig 2.** Mean absolute SHAP values of the original categorical features.

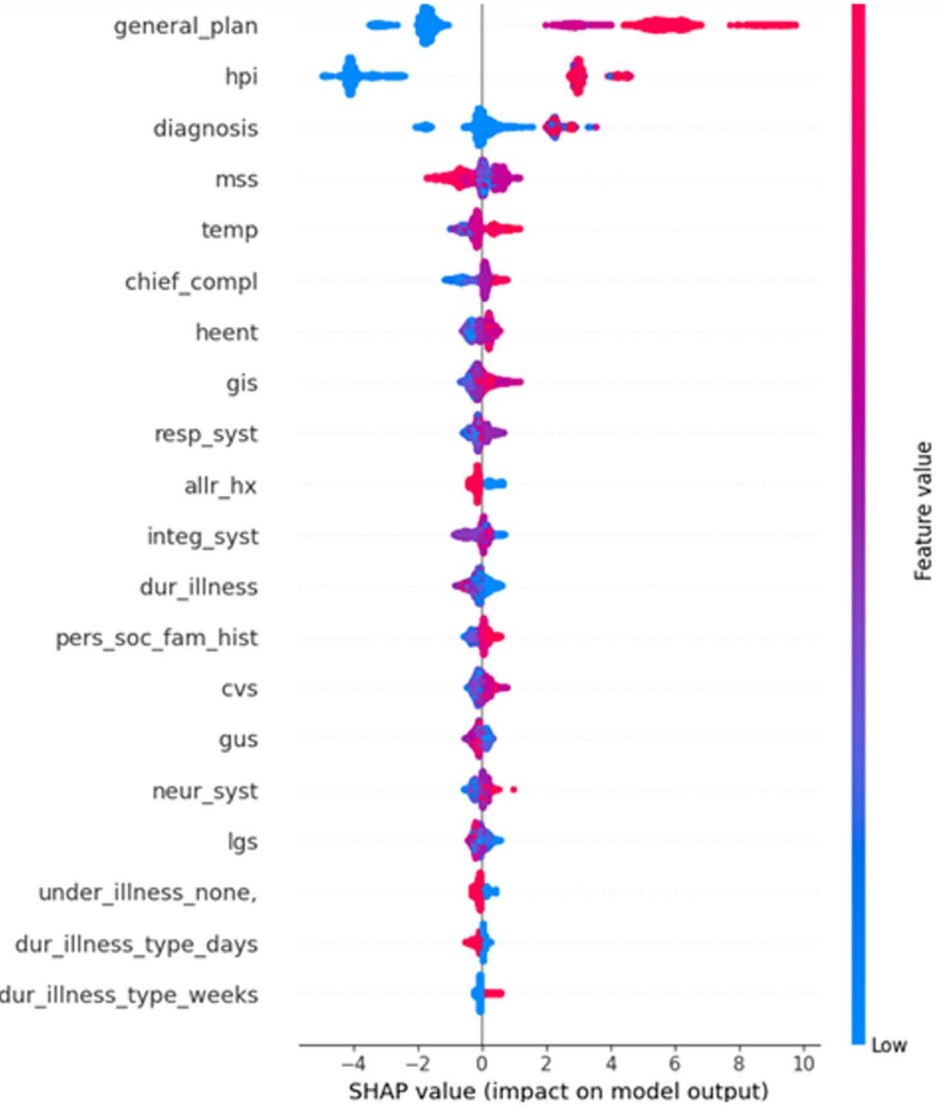

**Fig 3. SHAP summary plot for original categorical feature.**

This custom SHAP summary plot at Fig 4 displays the impact of the one-hot encoded features derived from the original 27 categorical variables. Since these features were expanded into 488 encoded variables, each bar here represents a specific encoded category

The plot at Fig 5 displays the SHAP value distribution for the one-hot encoded categories under the original categorical feature "general_plan." The X-axis represents the SHAP value, indicating how much each category contributes to the model's output. A positive SHAP value means that the presence of that category increases the predicted probability of the positive class (Label 1), whereas a negative SHAP value decreases it (increasing the probability of Label 0). SHAP values near zero suggest minimal influence.

The Y-axis lists the encoded categories, such as `general_plan_Other`, `FollowUp`, and Surgery. Each violin plot shows the distribution of SHAP values for a given category. The width of the violin at each point reflects the number of instances with that SHAP value, which helps us understand how consistently impact each category is. For example,

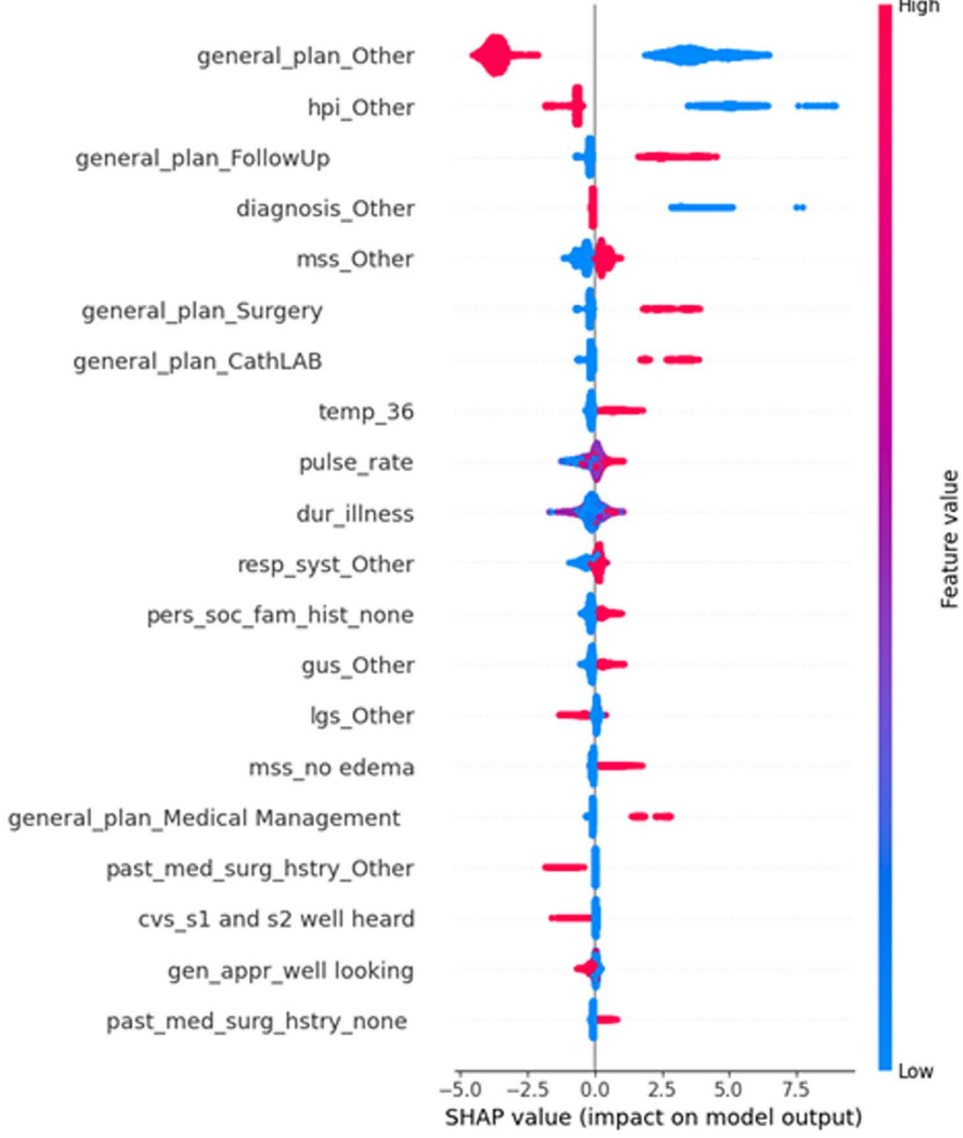

**Fig 4. SHAP values for all one hot encoded feature.**

`general_plan_Other` has a wide and symmetric distribution, indicating high variability—it can significantly increase or decrease predictions depending on the context. In contrast, `FollowUp` mostly pushes predictions toward Label 1, whereas surgery tends to reduce them.

The inner quartile markings within each violin (median, interquartile range, and overall range) provide further insight into where most values fall. These shapes help identify not only whether a category is influential but also the direction and consistency of its influence on model predictions.

## 4.5. Narrative explanations

The large language model (LLM) Gemini provides interpretable narratives that elucidate the model's decision-making ability by highlighting feature contributions to cardiovascular disease (CVD) risk prediction, as summarized in Fig 3. The

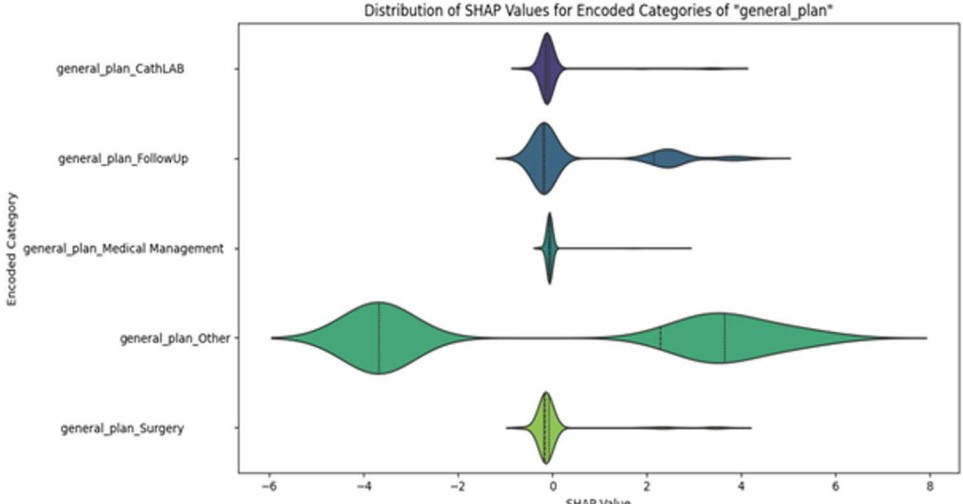

**Fig 5. SHAP value distribution of the encoded features of "General_Plan".**

dominant features—General Plan (52.6%) and History of Present Illness (HPI) (15.2%)—jointly contribute nearly 68% of the predictive power. These likely represent high-level care strategies and the current illness context. However, the prominence of the General Plan warrants careful auditing to prevent potential data leakage, whereas the HPI reflects clinically valid reasoning. Standardizing these inputs via structured clinical documentation is recommended to ensure data integrity.

Moderate contributors, such as the mean sum of squares (MSS, 4.3%), diagnosis (4.2%), and various patient histories, collectively add significant value (~23%), providing a comprehensive background and systemic context. These features align well with clinical diagnostic processes and highlight the importance of complete, integrated electronic health records to support accurate prediction.

Less influential features—such as respiratory and neurological systems, blood pressure, and chief complaint—show minimal individual contributions, potentially owing to inconsistent documentation or missing data. Despite their low model impact, these models remain clinically important; improving data capture and standardization could enhance future model performance. The narrative explanations thus offer valuable insight into feature relevance, guiding both model refinement and clinical data quality improvements.

## 5. Discussion

This study demonstrates the potential of combining the XGBoost algorithm with SHAP interpretability and large language model (LLM)-based narrative explanations to deliver highly accurate and transparent cardiovascular disease risk predictions. The results are consistent with those of prior machine learning studies on CVD prediction [8,9] while advancing the field by integrating real-world Ethiopian electronic medical record (EMR) data and providing interpretable narratives to support clinical decision-making.

### 5.1. Clinical implications

The developed framework offers a valuable decision-support tool for early CVD detection and is particularly suitable for resource-limited hospital settings. By enhancing clinician trust through transparent explanations and clear narratives, this approach facilitates adoption in clinical practice. Moreover, the system's design allows for potential integration into mobile health (mHealth) applications or EMR-based risk dashboards, broadening its usability.

### 5.2. Limitations and future work

While the XGBoost model demonstrated excellent predictive performance in this study, several limitations should be acknowledged. First, although XGBoost was prioritized for its efficiency, regularization capabilities, and ability to handle high-dimensional structured data, other machine learning models such as Random Forest, Support Vector Machines (SVM), and Deep Neural Networks (DNN) were not extensively tested. Future work should include systematic comparisons with these alternative models to explore potential improvements in predictive accuracy and interpretability.

Second, the study relied solely on EMR data from public hospitals under the Addis Ababa Health Bureau. External validation using independent datasets from other hospitals or regions is necessary to assess the generalizability and robustness of the model across different patient populations and healthcare settings. Incorporating such external datasets is planned for future studies to strengthen the model's applicability in broader clinical contexts.

Additionally, the current model does not integrate unstructured clinical data, such as physician notes or imaging reports, which may hold further predictive value. Future research should consider incorporating these sources to enhance the model's comprehensiveness. Finally, the large language model (Gemini) used for narrative generation has not been fine-tuned for the Ethiopian linguistic and clinical context, which may affect the clarity and cultural relevance of explanations. Fine-tuning or adopting locally adapted models could improve the interpretability and clinical usability of generated narratives.

## 6. Conclusion

We developed a highly accurate and interpretable cardiovascular disease prediction framework leveraging XGBoost, SHAP, and LLM-generated narratives on Ethiopian EMR data. This approach not only achieves strong predictive performance but also fosters clinician trust and usability through transparent explanations. Future research should prioritize prospective validation, the integration of unstructured clinical data, and the deployment of the system in real-time clinical environments via web and mobile platforms.

## Acknowledgments

The authors would like to express their sincere gratitude to the Addis Ababa Health Bureau for granting access to the anonymized electronic medical record (EMR) data used in this study. We acknowledge that the data were obtained under strict confidentiality and data protection policies, and were used solely for academic and non-commercial research purposes in full compliance with the bureau's ethical and regulatory requirements. Without their support, this research would not have been possible.

## Author contributions

**Data curation:** Kibrom Tewolde.

**Formal analysis:** Alemu Kumilachew Tegegnie.

**Investigation:** Alemu Kumilachew Tegegnie, Kibrom Tewolde.

**Methodology:** Alemu Kumilachew Tegegnie.

**Validation:** Alemu Kumilachew Tegegnie, Kibrom Tewolde.

**Visualization:** Alemu Kumilachew Tegegnie, Kibrom Tewolde.

**Writing – original draft:** Alemu Kumilachew Tegegnie, Kibrom Tewolde.

**Writing – review & editing:** Alemu Kumilachew Tegegnie, Kibrom Tewolde.

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
