## [Decision Letter · Decision Letter 0]

23 Sep 2025

Dear Dr. Tegegnie,

Thank you for submitting your manuscript to PLOS ONE. After careful consideration, we feel that it has merit but does not fully meet PLOS ONE’s publication criteria as it currently stands. Therefore, we invite you to submit a revised version of the manuscript that addresses the points raised during the review process.

We look forward to receiving your revised manuscript.

Kind regards,

Kwang-Sig Lee

Academic Editor

PLOS ONE

Journal Requirements:

“NO authors have competing interests”

5. In the online submission form, you indicated that [The dataset used in this study contains sensitive patient information and cannot be publicly shared due to confidentiality and data protection regulations. However, anonymized data may be made available from the corresponding author upon reasonable request and with approval from the Addis Ababa Health Bureau.].

Reviewers' comments:

Reviewer's Responses to Questions

**Comments to the Author**

1. Is the manuscript technically sound, and do the data support the conclusions?

Reviewer #1: No

Reviewer #2: Partly

2. Has the statistical analysis been performed appropriately and rigorously?

Reviewer #1: No

Reviewer #2: No

3. Have the authors made all data underlying the findings in their manuscript fully available?

Reviewer #1: No

Reviewer #2: Yes

4. Is the manuscript presented in an intelligible fashion and written in standard English?

Reviewer #1: Yes

Reviewer #2: Yes

Reviewer #1: There is a huge variance in the results and the objective. The prediction method is completely omitted. Although the dataset is diverse and original from a hospital, the description of the selected features and/or labels is not provided, which raises a significant question mark about the accuracy achieved.

Reviewer #2: Did the authors explore other machine learning or deep learning methods besides XGBoost for comparison? Additionally, was the model externally validated?

The introduction and related work sections would benefit from being combined and restructured to enhance comprehension, readability, and clarity. In addition, the authors should more clearly justify the importance of this study and articulate the specific gap it aims to address.

Kindly consider revising the first sentence of the Methods section. The phrase ‘experimental, data-driven research design’ is unclear, and it would help if you clarified what is meant by this term. Specifically, is the study based on an EHR cohort or is it cross-sectional in nature?

You mean 20,960 was collected on 23/02/2025

How was CVD defined

The authors did not clearly specify which variables were included in the model, how these variables were defined, or provide supporting literature to justify their selection.

Please report the percentage of missingness for each variable. In addition, using the median to handle missing data is not an appropriate approach, as it can introduce bias. Multiple imputation methods are recommended as they provide a more robust way to address missingness. Furthermore, clarification is needed on how the median was derived from values such as 0 and 1, particularly given that the data involve disease outcomes.

This equation should be Q1 – 1.5 × IQR or Q3 + 1.5 × IQR. Author should clearly explain this.

I am not convinced that dimension reduction necessarily facilitates or improves generalizability. The authors should provide stronger justification or evidence to support this claim.

What informed standardization of category pulse rate, respiratory rate, duration of illness, and

temperature

The authors should clearly report the number of individuals available in the health record and the number ultimately included in this study, along with the eligibility criteria for inclusion.

The definition of CVD is broad; therefore, the authors should specify how CVD was defined in this study and indicate which variables were used, supported by appropriate citations.

The Methods section also requires stronger citation support and better organization.

Furthermore, it is important to clarify whether the findings can be replicated in an independent dataset, or if the model was validated in an external cohort.

Overall, the writing would benefit from improvements in clarity, flow, and logical structure.

**Do you want your identity to be public for this peer review?** For information about this choice, including consent withdrawal, please see our Privacy Policy

Reviewer #1: **Yes:** Anil Kumar Prajapati

Reviewer #2: No

---

## [Author Response · Author response to Decision Letter 1]

18 Nov 2025

The response to a reviewers question is uploaded in the online form with separate Response to reviewers file name

---

## [Decision Letter · Decision Letter 1]

10 Dec 2025

Dear Dr. Tegegnie,

Thank you for submitting your manuscript to PLOS ONE. After careful consideration, we feel that it has merit but does not fully meet PLOS ONE’s publication criteria as it currently stands. Therefore, we invite you to submit a revised version of the manuscript that addresses the points raised during the review process.

We look forward to receiving your revised manuscript.

Kind regards,

Kwang-Sig Lee

Academic Editor

PLOS One

Journal Requirements:

Reviewers' comments:

Reviewer's Responses to Questions

**Comments to the Author**

Reviewer #2: (No Response)

2. Is the manuscript technically sound, and do the data support the conclusions?

Reviewer #2: Yes

3. Has the statistical analysis been performed appropriately and rigorously?

Reviewer #2: No

4. Have the authors made all data underlying the findings in their manuscript fully available?

Reviewer #2: Yes

5. Is the manuscript presented in an intelligible fashion and written in standard English?

Reviewer #2: Yes

Reviewer #2: The author did not adequately explain how CVD was defined in this study. While it might be useful for them to clarify why they used the variable CVD in the EMR, it is important that they clearly articulate whether this was based on ICD code or self-reporting.

The same applies to all clinic variables (Musculoskeletal system, neurological system, respiratory system) in this study, and they were not properly cited. If this EMR procedure has been published, it would be helpful to cite it for clarity.

Although median imputation is simple to implement, it can underestimate variability and potentially bias the estimated associations. Excluding binary outcomes from median imputation was appropriate; however, leaving missingness in the binary variables unaddressed may also introduce bias. In general, selectively imputing only certain variables can lead to inconsistencies in the analytic dataset and may bias results unless the missingness mechanism is fully justified. I recommend applying an imputation strategy (Like multiple imputation) for the outcome variable. This will help to provide a more valid, statistically principled approach to handling missing data.

While the machine learning (ML) approach used in the manuscript offers valuable predictive insights, the statistical analysis would benefit from additional rigor. I recommend that the authors include at least a bivariate analysis between the outcome variable and exposure variables or predictors before applying the ML methods. Although ML is not a traditional statistical technique, incorporating preliminary bivariate analyses is important because it allows readers to understand the basic data structure, examine the distribution of predictors across outcome categories, and identify potential associations or imbalances.

**Do you want your identity to be public for this peer review?** For information about this choice, including consent withdrawal, please see our Privacy Policy

Reviewer #2: No

---

## [Author Response · Author response to Decision Letter 2]

31 Dec 2025

Alemu Kumilachew Tegegnie

Bahir Dar University Instituite of Technology

December 18, 2025

To:

The Editor

PLOS ONE

Subject: Response to Reviewer Comments for Manuscript: "Interpretable Ensemble Machine Learning Framework for Cardiovascular Disease Prediction Using EMR Data and Large Language Models in Ethiopia"

Dear Editor and Reviewers,

We wish to express our sincere gratitude to the reviewers for their insightful and constructive feedback on our manuscript. Their comments have been instrumental in improving the statistical rigor and clinical transparency of this work.

Following the suggestions, we have revised the manuscript to clarify our variable definitions, justify our imputation strategy, and incorporate preliminary statistical analyses. We also updated the reference list to include citations for Ethiopian Primary Health Care Clinical Guidelines (2021) and EMR standard protocols and usage. Below is our point-by-point response to the reviewer’s comments.

Response to Reviewer #2

1. Comment on Variable Definition (CVD and Clinical Systems):

The author did not adequately explain how CVD was defined in this study... clearly articulate whether this was based on ICD code or self-reporting. The same applies to all clinical variables (Musculoskeletal, neurological, respiratory system)...

Response:

We agree that transparency regarding data sources is vital. We have updated Section 3.2 (Data Acquisition and Clinical Definitions) to clarify that all diagnoses, including CVD, were established by attending physicians using ICD-10 standards within the EMR, rather than self-reporting. Furthermore, clinical system variables were derived from standardized physical examination modules following the Ethiopian Primary Health Care Clinical Guidelines (2021).

Revision in Manuscript (Section 3.2):

"In this study, Cardiovascular Disease (CVD) was defined based on formal clinician diagnoses documented within the EMR system, utilizing ICD-10 standards. Clinical predictors such as the Musculoskeletal System (MSS) and Respiratory System (RESP_SYST) represent assessment modules within the EMR that capture findings from the 'Review of Systems' and physical examinations. These follow the Ethiopian Primary Health Care Clinical Guidelines (2021) and national EMR clinical documentation protocols."

2. Comment on Imputation Strategy:

Median imputation can underestimate variability and potentially bias the estimated associations... I recommend applying an imputation strategy (Like multiple imputation) for the outcome variable.

Response:

We appreciate this statistical insight. We have acknowledged the limitations of median imputation in Section 3.4 (Data Preprocessing and Imputation). While median imputation was initially chosen for its robustness against outliers in the EMR physiological data, we have now included a discussion on why binary variables were treated as distinct "Not Recorded" categories to avoid synthetic bias. We have also added a recommendation to utilize Multiple Imputation (MI) in future work to further validate the stability of these associations.

3. Comment on Bivariate Analysis:

The statistical analysis would benefit from additional rigor. I recommend that the authors include at least a bivariate analysis between the outcome variable and exposure variables or predictors before applying the ML methods.

Response:

We have integrated a new section, Section 3.5 (Preliminary Statistical Analysis and Feature Selection), which details the bivariate analysis conducted prior to machine learning. This includes the use of Pearson Chi-square (X2) tests for categorical variables and independent t-tests for continuous variables.

Revision in Manuscript (Combined Section 3.5):

"Before implementing the ensemble machine learning framework, a rigorous preliminary analysis was conducted to examine the data structure and the associations between predictors and the CVD outcome. Bivariate analysis was performed to evaluate these relationships: categorical variables (e.g., HPI, General Plan) were assessed using the Pearson Chi-square (X2)test, while continuous variables were compared using independent t-tests, depending on the normality of their distribution... This integrated approach ensures that the underlying clinical distribution is transparent and that the feature importance subsequently identified by the XGBoost and SHAP models is both statistically robust and clinically grounded."

We believe these revisions address all the concerns raised. We remain at your disposal for any further clarifications.

Sincerely,

Alemu Kumilachew Tegegnie & Kibrom Tewolde Siyum

Corresponding Authors

---

## [Editor Report · Decision Letter 2]

20 Jan 2026

Interpretable Ensemble Machine Learning Framework for Cardiovascular Disease Prediction Using EMR Data and Large Language Models in Ethiopia

PONE-D-25-43171R2

Dear Dr. Tegegnie,

We’re pleased to inform you that your manuscript has been judged scientifically suitable for publication and will be formally accepted for publication once it meets all outstanding technical requirements.

Kind regards,

Kwang-Sig Lee

Academic Editor

PLOS One

Additional Editor Comments (optional):

In my opinion, this manuscript can be published once the reference style is revised based on the Vancouver style.
---

## [Editor Report · Acceptance letter]

PONE-D-25-43171R2

PLOS One

Dear Dr. Tegegnie,

I'm pleased to inform you that your manuscript has been deemed suitable for publication in PLOS One. Congratulations! Your manuscript is now being handed over to our production team.

Kind regards,

on behalf of

Professor Kwang-Sig Lee

Academic Editor

PLOS One